# Depression and Its Associated Factors among Undergraduate Engineering Students: A Cross-Sectional Survey in Thailand

**DOI:** 10.3390/healthcare11162334

**Published:** 2023-08-18

**Authors:** Yuanyue Huang, Tinakon Wongpakaran, Nahathai Wongpakaran, Patraporn Bhatarasakoon, Preda Pichayapan, Shirley Worland

**Affiliations:** 1Graduate School, Chiang Mai University, Chiang Mai 50200, Thailand; yuanyue_huang@cmu.ac.th (Y.H.); nahathai.wongpakaran@cmu.ac.th (N.W.); patraporn.t@cmu.ac.th (P.B.); preda.p@cmu.ac.th (P.P.); shirley.worland@cmu.ac.th (S.W.); 2Department of Psychiatry, Faculty of Medicine, Chiang Mai University, Chiang Mai 50200, Thailand; 3Faculty of Nursing, Chiang Mai University, Chiang Mai 50200, Thailand; 4Faculty of Engineering, Chiang Mai University, Chiang Mai 50200, Thailand; 5Faculty of Social Sciences, Chiang Mai University, Chiang Mai 50200, Thailand

**Keywords:** depression, engineering undergraduates, romantic relationship, self-esteem, interpersonal difficulty, social skill, neuroticism

## Abstract

Background: Depression is a common mental health problem that can affect everyone at different stages of development. Though the prevalence rate of depression among university students is rising, exploration among engineering students is limited. The aim of the study was to examine the prevalence of depression and its associated factors of among engineering students in Thailand. Methods: A total of 346 students participated in this study. All completed the outcome inventory depression subscale (OI-D) to evaluate the level and prevalence of depression. Other variables assessed included social skills, learning styles, relationship satisfaction, interpersonal difficulty, alcohol use, internet addiction, and neuroticism. Correlation and regression analyses were applied to test the association between sociodemographic and psychosocial factors and depression. Results: Of the 346 students with the mean age of 20.25 (SD, 1.33), 52.31% were male. Based on the OI-D, 35.3% of participants exhibited symptoms indicative of major depression. Multiple regression showed that only neuroticism, interpersonal difficulties, social skills, and self-esteem appeared to be the significant predictors of depression. Conclusion: The prevalence of depression among engineering students in Thailand was unexpectedly high compared with the prevalence of depression among engineering students in some other countries. Intra- and interpersonal factors were found to be associated with depression. Further study on identifying these risk factors should be encouraged.

## 1. Introduction

Extensive research has been conducted on depression among university students, revealing a significant prevalence of depressive symptoms at 24.4% [1]. It is crucial to address these symptoms and emotions promptly, as untreated depression can potentially lead to various physical and psychological complications [2,3]. Suicidality represents the most severe manifestation of depression, with a systematic review involving nearly 90,000 college students in China indicating that individuals with depression had 2.2 times greater odds of developing suicidal ideation [4]. While mental health research among university students has predominantly focused on health science students, particularly medical students, recent systematic reviews have shed light on the prevalence of depression within this population. Notably, a review specifically targeting medical students found a depression prevalence of 41%, with several major risk factors identified. These risk factors included being female, being in the junior or preclinical years, exposure to COVID-19, academic stress, a history of psychiatric or physical disorders, economic troubles, fear of educational impairment, difficulties with online learning, fear of infection, feelings of loneliness, low physical activity, limited social support, problematic internet or smartphone use, and young age [5].

Additionally, culture can play a significant role in shaping mental health problems. Traditional East Asian culture, for example, places a strong emphasis on collectivism, in contrast to the individualism commonly found in the Western culture. While Asian culture may contain exhibitions of stigma around mental health and limited emotional expression, it also contains values of familial and social support [6,7]. Students with collectivistic culture backgrounds might face heightened academic and societal pressure. Furthermore, the expression of mental health problems, including depression, can be notably influenced by cultural factors [8,9,10].

Engineering students are gaining attention for comparative studies with medical students because both professions are stressful [11]. Like medical students, engineering students require long hours of studying, training, and practice. Engineering disciplines cover many subspecialties, including but not limited to civil engineering, electrical engineering, chemical engineering, etc. Most branch majors study general subjects such as algebra, statistics, chemistry, and physics. The study plans and study purposes are different between different branches, which leads to the different subjects of study. For example, civil engineering students study subjects including hydraulic engineering, structural engineering, materials, foundation engineering, etc. The main task of engineering education is to train students to gain sufficient knowledge and skills that meet the needs of society. Engineering graduates primarily proceed to the front line of industrial production, where they were engaged in design, manufacturing, operation, construction, research and development, and management. Engineering education varies due to different national conditions. The training focus also differs, but generally, most countries pay more attention to the practical operation ability of students, aiming to help them adapt to social life in the shortest time after graduation [12,13]. Students are overburdened with a massive amount of information, having a limited amount of time to memorize all the information studied. Information overload creates a feeling of disappointment, inability to handle the knowledge and increased incidence of errors which ultimately break the stability of the student’s wellness and result in illness [14]. Researchers found that depression among engineering students was associated with internet addiction and the frequency of internet use. High frequency of internet use and internet addiction are positively correlated with depression among engineering students [15,16], social skills are negatively correlated with clinical depression [17], and alcohol use is a strong predictor of depression. Individuals who start using alcohol while underage are more likely to become addicted than individuals who started at age 20 or over [18]. Particularly, alcohol use was found in two thirds of males according to the Thai National Mental Health Survey (2013), which consequently led to psychotic symptoms, intermittent explosive disorder, and panic disorder [18].

Different disciplines may require different learning styles. A study revealed that students with visual and kinesthetic styles of learning are likely to develop depression [19]. In comparison to medical students who had the quadmodal VARK (Visual, Aural, Read/Write, Kinesthetic) mode as the most preferred style [20], approximately 25% of students preferred a multi-modal learning style among engineering students [21]. No associations between VARK modal preferences and depression were observed among medical students.

Although limited in number, research studies on psychosocial factors associated with depression among engineering students have gained attention. Personality traits, such as neuroticism, interpersonal problems, and self-esteem, have been extensively studied and found to have significant influence on depression [14,22,23,24,25,26]. Furthermore, relationship issues hold particular significance for young individuals during this stage of life. Previous studies have demonstrated that relationship satisfaction is associated with elevated stress levels, academic difficulties, and depression among university students [17,27,28]. Examining the interplay between relationship satisfaction and depression within the context of engineering students could provide valuable insights into the unique challenges they face.

## 2. The Present Study

The authors aimed to examine the prevalence of depression among undergraduate engineering students in Thailand and identify potential associated factors. The investigation encompassed sociodemographic data, learning styles, social skills, and various psychosocial factors. Building upon the existing literature on university students, we anticipated that depression would also be prevalent among engineering students. Our hypothesis focused on certain psychosocial variables, such as low self-esteem and neuroticism, as significant predictors of depression in this population. Additionally, we aimed to compare alcohol use prevalence between engineering students and health science students. Furthermore, we explored the distinctive learning styles preferred by engineering students, although we did not formulate a specific hypothesis regarding the association between learning styles and depression. Ultimately, this study aimed to contribute to the understanding of mental health among engineering students and provide valuable insights for developing targeted interventions and support programs.

## 3. Materials and Methods

### 3.1. Study Design

This study is a quantitative cross-sectional descriptive online survey. The study was approved by the ethics committee of the Faculty of Medicine, Chiang Mai University. Study code: PSY-2565-09146, certification number 340/2022 effective 5 October 2022. 

### 3.2. Sample Size Calculation

The study required a sample size of 317. According to the Statistical Thailand [29], the number of engineering students in Thailand in 2018 was 30,346. With the calculation formula [30], the estimated number is 345 people in total.

### 3.3. Participants and Setting

Participants needed to be Thai undergraduates aged 18–25 years, and they could be pursuing any major related to engineering. Participants were asked to be fluent in Thai and be able to access the internet. Participants could be from any region and any university in Thailand, nationally. The exclusion criteria included those diagnosed with schizophrenia, bipolar disorder, drug or alcohol use disorder. Sociodemographic data, including academic information such as grade point average (GPA), as well as physical/mental illness and social relationships, including current romantic relationships, were collected.

### 3.4. Measurements

#### 3.4.1. Social Skills Inventory (SSI)

The Social Skills Inventory (SSI) includes six scales used to measure communication skills in both emotional (non-verbal) and social (verbal) dimensions and to assess basic social skills [31]. SSI consists of 30 items; the total scores for data analysis were summed by these 30 items. An example sample of an SSI item is ‘I am able to conceal my true feelings from just about anyone’. The Thai version of SSI showed [32] a Cronbach’s alpha of 0.88 [17]. The Cronbach’s alpha of SSI in the current study was 0.853.

#### 3.4.2. Relationship Assessment Scale (RAS)

The RAS measures general relationship satisfaction [33]. It has 7 items (e.g., How well does your partner meet your needs? In general, how satisfied are you with your relationship? How good is your relationship compared to most?). Respondents answer each item using a 5-point scale ranging from 1 (low satisfaction) to 5 (high satisfaction) [33]. The total scores for data analysis were summed by 7 items. Individual total scores on the RAS can range from 7 to 35, higher scores mean that the individual experiences higher satisfaction with the relationship. The Cronbach’s alpha score of relationship assessment scale for parents, friends and romantic partners was 0.89, 0.87 and 0.90, respectively [34]. The Cronbach’s alpha of RAS in the current study was 0.834.

#### 3.4.3. Visual Aural Read/Write Kinesthetic (VARK)

The VARK questionnaire, which can be used to assess individual learning preference, includes 16 multiple choice questions (e.g., I need to find the way to a shop that a friend has recommended. I would: (a) find out where the shop is in relation to somewhere I know; (b) Ask my friend to tell me the directions; (c) Write down the street direction I need to remember; (d) Use a map.). All choices correspond to the four sensory modalities which are measured by VARK [19]. The total scores of Visual, Aural, Read/write and Kinesthetic were summed, respectively, for data analysis. The Cronbach’s alpha scores of the learning styles were 0.85, 0.82, 0.84, and 0.77 for Visual, Aural, Read/write, and Kinesthetic styles, respectively [35]. The Thai version of VARK showed the Cronbach’s alpha of VARK in the current study of 0.666, 0.576, 0.655, and 0.648 for Visual, Aural, Read/write, and Kinesthetic learning styles, respectively [36].

#### 3.4.4. Outcome Inventory-21(OI-21)

The OI-21 measures anxiety, depression, somatization, and interpersonal difficulty. It is a 21-item questionnaire with a 5-Likert-type response scale, from ‘0’, not at all, to ‘4’, almost always. The Cronbach’s alpha score of OI-21 was 0.92 [37]. The Cronbach’s alpha of OI-21 in the current study was 0.915. In the present study, only the subscales of depression and interpersonal difficulty were used.

Outcome Inventory Interpersonal difficulty (OI-I) measures interpersonal problems, specifically the socially inhibited style. This subscale has four items. The samples of the OI-I items include the following items: ‘I do not get along with others’; ‘I feel uncomfortable with people that are not family’. Individual scores on the OI-I range from 0 to 16, with higher scores indicating higher interpersonal problems. The total scores of OI-I for data analysis were summed by 4 items. The Cronbach’s alpha score of OI-I was 0.83 [37]. The Cronbach’s alpha of OI-I in the current study was 0.796.

Outcome Inventory Depression (OI-D) measures depressive symptoms. This subscale has five items. The samples of the OI-D items include the following items: ‘I believed that I cannot have a happy life—as others do’. The total scores if OI-D for data analysis were summed by 5 items. Individual total scores on the OI-D range from 0 to 20, with higher scores indicating a higher level of depression. The Cronbach’s alpha score of OI-D was 0.83 [37]. The Cronbach’s alpha of OI-D in the current study was 0.847. The cut-off score of 7 was used to determine major depression [36].

#### 3.4.5. Internet Addiction Test (IAT)

The IAT, developed by Young, measures mild, moderate, or severe levels of internet addiction [38]. It has 20 questions (e.g., How often do you find that you stay online longer than you intended?), and participants rate themselves on a scale of 0 (not applicable) to 5 (always) based on how often the questions apply to them. IAT has been widely used in global academic studies. Its effectiveness and practicality in diagnosing internet addiction have been confirmed in previous studies [39]. The total scores of IAT for data analysis were summed by 20 items. Individual scores on the IAT range from 0 to 100. Higher scores mean individuals have higher levels of internet addiction. The Thai version demonstrated that the Cronbach’s alpha score of IAT was 0.89 [40]. The Cronbach’s alpha of IAT in the current study was 0.919.

#### 3.4.6. The Revised Thai Version of Rosenberg Self-Esteem Scale (RSES)

The Rosenberg Self-esteem Scale (RSES), developed by Rosenberg [41] was initially used to assess the overall feelings of self-worth in adolescents. The scale is composed of 6 items (e.g., On the whole, I am satisfied with myself; I feel that I have a number of good qualities) with positive scores and 4 items with negative scores (e.g., At times I think I am no good at all; I feel I do not have much to be proud of). It is a 4-rating scale ranging from ‘1’, strongly disagree, to ‘4’, strongly agree. The total scores of RSES for data analysis were summed by 10 items. Individual total scores on the RSES range from 0 to 40, with higher scores indicating a higher level of self-esteem. The Cronbach’s alpha score of the Thai version of RSES was 0.87 [42]. The Cronbach’s alpha of RSES in the current study was 0.91.

#### 3.4.7. Neuroticism Inventory (NI)

The NI is a dimensional measure of the neuroticism personality trait based on Eysenck’s five-factor model [43]. The NI, developed by Wongpakaran and Wongpakaran [44], has 15 items in total, with a 4-Likert-type scale, from 0 (never like me) to 4 (like me the most) (e.g., I tend to brood over things; I often feel stressed). The total scores of NI for data analysis were summed by 15 items. Individual total scores on the NI range from 0 to 60, with a higher score reflecting a higher level of neuroticism. The Cronbach’s alpha score of NI was found to be 0.83 [44]. The Cronbach’s alpha of NI in the current study was 0.92.

### 3.5. Statistical Analysis

The data analysis of socio-demographic characteristics and the prevalence of depression used descriptive analysis to calculate frequency, percentage, mean and standard deviation (SD). The prevalence of depression was calculated by using the cut-off score for OI-D. We employed ANOVA and *t*-test to analyze the difference of the continuous scores across the group. The chi-square test was used to analyze the association between categorical variables across groups. 

Pearson correlation coefficient was employed to assess the relationships among variables with continuous scores such as relationship satisfaction, self-esteem scores and neuroticism scores. 

Variables with significant correlation with depression score (*p* < 0.05) were included in multiple linear regression model. The beta values derived from the analysis results were used to elucidate the nature of the relationship between each independent variable and depression, whether positive or negative. The normality of the residual were evaluated to confirm that the assumption of regression model was met. Additionally, the *p* values accompanying these coefficients indicated the statistical significance of the relationships. Specifically, a *p* value below 0.05 signified that a particular factor serves as a robust predictor of depression. IBM SPSS version 22 was used for all analyses.

## 4. Results

### 4.1. Sociodemographic and Clinical Variables

In this study, a total of 346 engineering undergraduates from Thailand filled out the questionnaire. The participants included 181 males, 165 females. The male participants had an average age of 20.19 (SD = 1.35), while the female participants had an average age of 20.30 (SD = 1.30). No difference in age between sexes was observed. Most participants were Buddhists (84.7%). Regarding the participants’ engineering majors, 67 students were majoring in Electrical Engineering, accounting for 19.4% of the total. Civil Engineering was the major of 50 students (14.5%), followed by Mechanical Engineering (42 students, 12.2%), Industrial Engineering (40 students, 11.6%), and Chemical Engineering (30 students, 8.7%).

The overall mean GPA was 2.90 (SD = 0.55, n = 285); the male:female distribution was 2.92 (SD = 0.59) and 2.85 (SD = 0.51), respectively. No difference in GPA between sexes was observed. Most participants reported having no physical illnesses (93.4%) and no mental illnesses (97.9%). Among all, 60.3% reported alcohol consumption (31.0% male, 29.3% female). Most of them reported drinking alcohol once a week (75.8%). Only 6.6% admitted that alcohol significantly affected their life.

Information regarding the participants’ monthly family income and monthly allowance was collected. Among the participants, 129 (37.3%) reported a monthly family income between 25,000 and 50,000 baht. In terms of monthly allowance, 155 (45.1%) participants received between 5000 and 10,000 baht per month (Table 1).

Romantic relationships were explored; 148 (42.9%) participants reported being in a romantic relationship. Of those in a relationship, 82 reported seeing their partner at least once a day (56.2%), while 2 reported seeing their partner every six to seven months (1.4%). 

### 4.2. Learning Styles Results

Out of a total of 340 valid samples, 92 individuals (27.06%) exhibited a unimodal learning style (V or A or R or K), 42 individuals (12.35%) exhibited a bimodal learning style (e.g., VA, RK), 60 individuals (17.65%) exhibited a trimodal learning style (e.g., VAR, ARK), and 146 individuals (42.94%) exhibited a quadmodal learning style (VARK). No significant difference in depression scores between VARK modalities was observed. 

### 4.3. Pearson Correlation, Linear Regression between Psychosocial Variables and Depression

Pearson correlation analysis revealed that practicing Buddhism, being in a romantic relationship, and experiencing relationship satisfaction were negatively associated with depression (*p* < 0.01), whereas history of mental illness, neuroticism, internet addition, interpersonal difficulties, and currently being in a relationship were positively associated with depression (Table 2). 

Multiple linear regression analysis results revealed that self-esteem, neuroticism, interpersonal difficulty, and social skills significantly predicted depression score (*F* (10, 206) = 31.063, *p* < 0.001). This model explained a 58.2% variance of depression (Table 3).

## 5. Discussion

The aim of this study was to investigate the prevalence of depression among Thai engineering undergraduate students and examine the sociodemographic and psychosocial factors associated with depression, as well as the predictive factors for depression within the target population. The results of this study shed light on the relationship between psychosocial factors such as neuroticism, interpersonal difficulty, social skills and self-esteem, which significantly predicted depression. 

In this study, we found that the prevalence of depression among all 346 Thai undergraduate engineering students was 35.3%. These findings are consistent with those of previous research on the mental health of university students. A study reported a similar prevalence of depression (27.1%) among university students in a different country [45]. The higher prevalence rates of depression (35.3%) among engineering students in this study among Thai engineering undergraduates compared with the prevalence of depression in two Turkish papers which conducted a study among Turkish engineering students (20.4% and 26.2%, respectively) highlight the importance of addressing mental health issues in this specific population [46,47]. 

Despite the high prevalence rate, determining the extent of student access to healthcare is challenging. Several factors, including cultural stigmatization, contribute to this difficulty [48]. In many East Asian cultures students facing mental health issues may be reluctant to seek help or openly discuss their struggles.

The academic demands, intense competition, and social pressures experienced by engineering students may contribute to their elevated risk for mental health problems. The variation in the prevalence rates between the current study and the mentioned study could be attributed to the utilization of different measurement tools. In the present study, OI-Depression was employed, whereas Bayram and Bilgel’s research utilized the Depression Anxiety and Stress Scale (DASS-42).

As anticipated, several psychosocial variables were found to be associated with depression, including high levels of neuroticism, low self-esteem, elevated interpersonal difficulties, and lower level of social skills. These variables seem to be related to each other. Without question, low self-esteem is related to depression, regardless of age and population. The vulnerability model demonstrates that low self-esteem significantly influences depression, even though depression influences self-esteem as well. A metanalysis study has supported the former hypothesis [49]. 

It is to be noted that the interpersonal difficulty measured in this study is a socially inhibited interpersonal style. The fact that interpersonal difficulty predicted depression among Thai engineering undergraduates could be because interpersonal difficulties may lead to feelings of isolation and loneliness [50]. The nature of engineering education itself often emphasizes individual problem-solving and technical skills, which may not provide ample opportunities for developing strong interpersonal skills. Those with interpersonal difficulties may not have sufficient opportunity to cultivate more interpersonal skills. As a result, engineering students may struggle with effective communication, collaboration, and building supportive social networks, which are important protective factors against depression. Similarly, social skills follow a similar pattern. Numerous studies have found a correlation between a lack of social skills and depression [51,52,53]. Individuals who struggle with interpersonal communication often exhibit deficits in social skills. Cultural factors may also contribute to limited emotional expression and socially inhibited interpersonal behaviors, which can impact social skills and eventually lead to depression. This effect becomes particularly notable during the COVID-19 pandemic, as social isolation exacerbates the situation [54,55]. Moreover, researchers have identified loneliness as a potential mediator between social skills and depression [56], possibly operating through the pathway of self-esteem [57]. In a longitudinal study, it was discovered that fostering social skills in individuals may play a crucial role in preventing adolescent depression. This is because an improvement in social skills contributes to increased responsibility and self-control, which, in turn, corresponds to a slower rise in depression [53].

Based on previous research, it was anticipated that neuroticism would emerge as a significant predictor of depression, aligning with findings from similar studies [24,44,58,59,60,61]. Neurotic individuals tend to be more emotionally sensitive and reactive to stressful situations. Engineering undergraduates often face high levels of academic pressure, including demanding coursework, exams, and deadlines. The combination of neuroticism and these stressors may lead to heightened emotional vulnerability, making them more susceptible to experiencing depressive symptoms [62]. They may tend to focus on negative aspects of situations, exaggerate their significance, and have difficulties in finding positive interpretations. These cognitive biases can amplify negative emotions and thoughts, contributing to the development of depression. Like self-esteem, neuroticism is widely accepted to be a predictor for depression [63,64,65].

Regarding learning style, similar to other related studies, multimodal learning according to the VARK model was preferred among students. Among the engineering students in this study, 42.94% exhibited a quadmodal learning style, which was close to the percentage observed in medical students (43.57%). It is noteworthy to compare the depression scores between the study by Kunanitthaworn and colleagues and the current study, as both studies used the same depression measurement (OI-D) and included Thai samples [66]. Medical students reported higher levels of depression compared to engineering students. Similar to the medical students, no significant difference in depression based on the VARK style among these engineering students. This suggests that the relationship between learning style and depression is less significant when compared to that of the intra- and interpersonal factors. 

Furthermore, the study found that other factors such as relationship satisfaction and romantic relationships were weak and non-significant predictors. Being in a relationship was associated with decreased levels of depression. However, the quality of relationships seems to be more important. Romantic relationships often provide individuals with emotional support, understanding, and companionship. Having a supportive and loving partner can create a sense of belonging [67], reduce feelings of loneliness, and serve as a buffer against stressors, all of which may contribute to lower levels of depression. However, the significance of social skills in predicting depression was overshadowed by other influential factors. 

Similarly, the use of alcohol did not emerge as a predictor of depression. In Thai society, socializing and bonding often involves alcohol consumption, especially in certain social contexts such as celebrations or gatherings with peers. Engineering undergraduates may feel compelled to conform to these cultural norms and engage in alcohol consumption [46]. Research has shown that students who tend to avoid seeking help for depressive symptoms potentially take up drinking as a coping strategy [68]. However, other investigators found no correlation between alcohol and depression [69]. The current study confirmed this, as over 60 percent of participants consumed alcohol. Even though 14% reported having many problems with their lives due to alcohol, it was not a predictor for depression. Therefore, it is possible that engineering students tend to use alcohol for social drinking rather than as a coping strategy. The same is true for internet addiction. It can serve as a form of escapism, allowing individuals the retreat from real-life problems or difficulties they may be facing. This can lead to social isolation and a lack of face-to-face interactions, which are important for maintaining emotional well-being [70]. Without adequate social support, individuals may be more prone to developing depressive symptoms [71]. Even though the internet use problem was found to be related to depression, it was overruled by other important factors. Like alcohol consumption, excessive internet use can be viewed as a coping mechanism. The behavior itself is not indicative of depression.

## 6. Implications and Future Research

These findings offer valuable insights for universities, highlighting the importance of proactively implementing focused educational initiatives to address potential psychological challenges among engineering students. For instance, using screening tools to assess personality traits like neuroticism and interpersonal difficulties can help identify individuals at risk for depression. This identification can enable targeted interventions and support for at-risk students. Social skill training should be promoted, especially for those who lack interpersonal skills. A more comprehensive approach should be established to enhance early detection and intervention for mental health problems among university students [72]. Universities should implement prevention and early intervention programs through their healthcare facilities. These programs could focus on developing interpersonal and social skills among students, as well as initiatives that foster social connectedness and bolster self-esteem and self-efficacy. Early detection is crucial in identifying mental health issues before they escalate, and universities play a vital role in creating a supportive environment for students. By participating in proactive programs that address various aspects of mental well-being, such as social skills, social support, and self-confidence, students can be better equipped to cope with the challenges they face during their academic journey. These initiatives could involve workshops, counseling services, peer support groups, and awareness campaigns to raise mental health awareness on campus.

In future research, it is recommended to incorporate positive mental health variables such as character strength or inner strength. This would allow investigation into how these positive factors might serve as buffers against mental health problems like depression, particularly for individuals who have been identified as at-risk through positive screening. Exploring the protective effects of positive factors can contribute to a more comprehensive understanding of mental health and inform the development of effective preventive strategies and support systems. Conducting longitudinal studies can help establish the causal relationship between these predictive factors and depression. Long-term investigations can provide valuable insights into how changes in these strong predictive factors over time affect the development and course of depression. 

## 7. Limitation

It is essential to acknowledge that this study has several limitations. Firstly, the participants included in the study were exclusively Thai undergraduate engineering students. This sample specificity raises a concern regarding the generalizability of the findings to other populations or cultural contexts. As this study was conducted on Thai students, it may not be applicable to engineering students from diverse cultural backgrounds. To enhance the external validity of the results, future research should consider expanding the participant pool to include a more diverse range of individuals.

Secondly, the study relied on self-report measures as the primary data collection method. Self-report measures are susceptible to biases such as social desirability or recall biases, which could affect the accuracy and reliability of the reported information. Moreover, self-report measures may not capture the entire spectrum of mental health disorders and may be limited in their ability to provide a comprehensive assessment. Incorporating additional objective measures or clinician-administered assessments in future studies could provide a more comprehensive understanding of mental health outcomes. Lastly, the study is a cross-sectional study. It is difficult to determine a causal relationship between the outcome and predictive variables; thus, a longitudinal study is considered in the future study.

## 8. Conclusions

The study findings revealed a notable prevalence of depression among undergraduate engineering students in Thailand. The high rates of depression underscore the importance of addressing mental health concerns within this population. The unique combination of academic demands, intense competition, and social pressures experienced by engineering students contributes significantly to these elevated rates. Notably, the study established a significant correlation between psychosocial variables and depression in this group.

Specifically, individual (intra-personal) and interpersonal factors such as self-esteem, personality traits (e.g., neuroticism), interpersonal difficulties, and social skills emerged as robust predictors of depression, aligning with findings from related research. Furthermore, although internet addiction and alcohol consumption were prevalent among undergraduate engineering students in Thailand, they did not demonstrate themselves as essential predictors for depression.

Thai engineering students may share similar subjects and study environments with foreign students, but they could be influenced by the culture of collectivism. This cultural aspect can impact their mental health, particularly leading to socially inhibited interpersonal behaviors. These behaviors, in turn, may make individuals more susceptible to depression, especially when facing academic or relationship pressures.

These findings carry practical implications for universities, highlighting the necessity for targeted educational initiatives to prevent potential psychological issues among students. By recognizing the influence of psychosocial factors on depression, universities can implement interventions and support systems tailored to the unique needs of engineering students. Such initiatives can foster positive mental health and well-being within this student population.

## Figures and Tables

**Table 1 healthcare-11-02334-t001:** Sociodemographic characteristics of the participants.

Variables		Male (N = 181)	Female (N = 165)	Total (N = 346)	Test Difference
		Mean ± SD	Mean ± SD	Mean ± SD	
Age (18–25)		20.19 ± 1.35	20.30 ± 1.30	20.25 ± 1.33	t (338) = −0.78, *p* = 0.438
Religion	Buddhism	149 (43.1%)	144 (41.6%)	293 (84.7%)	Χ^2^(3) = 4.09, *p* = 0.252
	Christianity	5 (1.4%)	1 (0.3%)	6 (1.7%)	
	Islam	7 (2.0%)	8 (2.3%)	15 (4.3%)	
	No religion	20 (5.8%)	12 (3.5%)	32 (9.2%)	
Major	Mechanical Engineering	26 (7.5%)	16 (4.6%)	42 (12.2%)	Χ^2^(9) = 36.91, *p* < 0.001
	Electrical Engineering	46 (13.1%)	21 (6.1%)	67 (19.4%)	
	Civil Engineering	26 (7.5%)	24 (7.0%)	50 (14.5%)	
	Computer Engineering	17 (4.9%)	11 (3.2%)	28 (8.1%)	
	Chemical Engineering	11 (3.2%)	19 (5.5%)	30 (8.7%)	
	Biomedical Engineering	7 (2.0%)	10 (2.9%)	17 (4.9%)	
	General Engineering	11 (3.2%)	6 (1.7%)	17 (4.9%)	
	Industrial Engineering	11 (3.2%)	29 (8.4%)	40 (11.6%)	
	Environmental Engineering	3 (0.9%)	16 (4.6%)	19 (5.5%)	
	Others *	23 (6.7%)	12 (3.5%)	35 (10.1%)	
Marriage status	Single	176 (51.2%)	162 (47.1%)	338 (98.3%)	Χ^2^(1) = 0.01, *p* = 0.920
	Married	3 (0.9%)	3 (0.9%)	6 (1.7%)	
History of Physical illness	No	160 (48.2%)	150 (45.2%)	310 (93.4%)	Χ^2^(1) = 0.07, *p* = 0.790
	Yes	12 (3.6%)	10 (3.0%)	22 (6.6%)	
History of Mental illness	No	174 (51.6%)	156 (46.3%)	330 (97.9%)	Χ^2^(1) = 0.27, *p* = 0.605
	Yes	3 (0.9%)	4 (1.2%)	7 (2.1%)	
Drinking alcohol	Yes	107 (31.0%)	101 (29.3%)	208 (60.3%)	Χ^2^(1) = 0.22, *p* = 0.640
	No	74 (21.4%)	63 (18.3%)	137 (39.7%)	
Family monthly income	Lower than 25,000 THB	51 (14.7%)	48 (13.9%)	99 (28.6%)	Χ^2^(4) = 8.60, *p* = 0.072
	25,000–50,000 THB	60 (17.3%)	69 (19.9%)	129 (37.3%)	
	50,001–75,000 THB	33 (9.5%)	27 (7.8%)	60 (17.3%)	
	75,001–100,000 THB	13 (3.8%)	13 (3.8%)	26 (7.5%)	
	Over 100,000 THB	24 (6.9%)	8 (2.3%)	32 (9.2%)	
Monthly allowance	Lower than 5000 THB	66 (19.2%)	65 (18.9%)	131 (38.1%)	Χ^2^(4) = 4.37, *p* = 0.358
	5000–10,000 THB	81 (23.5%)	74 (21.5%)	155 (45.1%)	
	10,001–15,000 THB	18 (5.2%)	21 (6.1%)	39 (11.3%)	
	15,001–20,000 THB	7 (2.0%)	2 (0.6%)	9 (2.6%)	
	Over 20,000 THB	7 (2.0%)	3 (0.9%)	10 (2.9%)	
Latest semester GPA		2.95 ± 0.63	2.82 ± 0.61	2.91 ± 0.63	t (329) = 2.91, *p* = 0.004
Prevalence of depression		67 (19.5%)	55 (16.0%)	122 (35.3%)	X^2^ (1) = 0.57, *p* = 0.499

* Others include Aerospace and Aeronautical Engineering, Material Engineering, System Engineering, Electronic and Telecommunication Engineering, Mining Engineering, Mechatronics Engineering, Robots and Artificial Intelligence Engineering, Nano Engineering, Integrated Engineering, Software Engineering, Automotive Engineering, Agriculture Engineering, Logistic Engineering, Manufacturing Engineering, No limited branch.

**Table 2 healthcare-11-02334-t002:** Pearson Correlation between psychosocial variables and depression.

Variables	Depression
Age	−0.001
Sex (Male vs. Female)	−0.016
Grade point average	−0.130 *
Buddhism (vs. non-Buddhism)	−0.269 **
Relationship satisfaction	−0.210 **
Rosenberg self-esteem	−0.698 **
Neuroticism	0.563 **
Social skills	−0.238 **
Internet addition test	0.317 **
Interpersonal difficulties	0.550 **
History of mental illness (Yes vs. No)	0.179 **
Current romantic relationship (Yes vs. No)	0.188 **

** < 0.01; * < 0.05.

**Table 3 healthcare-11-02334-t003:** Multiple linear Regression for Depression.

	B	SE	Beta	*t*	*p*-Value	95.0% Confidence Interval for B	
						Lower Bound	Upper Bound
(Constant)	10.664	3.483		3.062	0.002	3.798	17.530
Relationship satisfaction	−0.037	0.039	−0.050	−0.929	0.354	−0.114	0.041
Rosenberg self-esteem	−0.418	0.044	−0.530	−9.573	0.000	−0.504	−0.332
Neuroticism	0.065	0.029	0.140	2.234	0.027	0.008	0.123
Interpersonal difficulty	0.319	0.083	0.263	3.828	0.000	0.155	0.484
Buddhism	0.285	0.546	0.024	0.521	0.603	−0.792	1.362
Internet addition test	0.027	0.015	0.097	1.838	0.068	−0.002	0.057
Current romantic relationship	0.121	0.401	0.015	0.302	0.763	−0.669	0.911
Past Mental illness	−1.895	1.892	−0.046	−1.001	0.318	−5.626	1.836
GPA	−0.244	0.338	−0.033	−0.722	0.471	−0.910	0.422
Social skills	0.048	0.021	0.131	2.313	0.022	0.007	0.089

Adjusted r square = 0.582, SE = Standard error, B = unstandardized coefficient, beta = standardized coefficient, GPA = grade point average.

## Data Availability

The datasets used and/or analyzed during the current study are available from the corresponding author upon reasonable request.

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
