# Peer review of "Depression and Its Associated Factors among Undergraduate Engineering Students: A Cross-Sectional Survey in Thailand"

_healthcare, 2023, doi:10.3390/healthcare11162334_

Round 1

Reviewer 1 Report

Authors should specify the following information: 

- unify in their text the formulation of the objective of their study.

 - specify whether the students are from Thailand's origin or are at the Thai University.

- Specify the course they are taking in their studies. 

- Take care of the orderly numbering of section 2.4 Measurements.

- To deepen the implications of the university in health and social welfare programmes. See 

Lázaro-Pérez, C.;Gómez, P.M.; Martínez-López, J.Á.;Gómez-Galán, J. Predictive Factors ofSuicidal Ideation in SpanishUniversity Students: A Health,Preventive, Social, and CulturalApproach. J. Clin. Med. 2023, 12, 1207.https://doi.org/10.3390/jcm12031207

Author Response

Dear Editor,

We are appreciated the editor and reviewers that provided us valuable comments. Please see below our point-by-point response to each comment.   Besides, all the manuscript has been checked and corrected some minor errors and mistakes. The revised parts are in color or highlighted.

Reviewer Comments:

Reviewer 1
1.1 unify in their text the formulation of the objective of their study.

Response. Thank you for your suggestion. We have added the objective of participants’ study, contents were added in the “Present study” section as followed.

The objective of this study was to examine the prevalence of depression among undergraduate engineering students in Thailand and identify potential associated factors. The investigation encompassed sociodemographic data, learning styles, social skills, and various psychosocial factors. Building upon existing literature on university students, we anticipated that depression would also be prevalent among engineering students. Our hypothesis focused on certain psychosocial variables, such as low self-esteem and neuroticism, as significant predictors of depression in this population. Additionally, we aimed to compare alcohol use prevalence between engineering students and health science students. Furthermore, we explored the distinctive learning styles preferred by engineering students, though we did not formulate a specific hypothesis regarding the association between learning styles and depression. Ultimately, this study aimed to contribute to the understanding of mental health among engineering students and provide valuable insights for developing targeted interventions and support programs.

1.2 specify whether the students are from Thailand's origin or are at the Thai University.

Response: Thank you for your suggestion. We did not set a demographic item to ask whether participants are from Thailand’ origin or not, because we set one of the inclusions is participants need to be a Thai people. This study conducted in Thailand nationally. We added some information in ‘Participants and setting’ section based on your suggestion to make it clearer, added information as followed.

‘Participants could from any regions and any universities in Thailand nationally.’

1.3 Specify the course they are taking in their studies.

Response: Thank you for your suggestion, we added some additional information in the second paragraph of ‘Introduction’ section to make views clearer to understand engineering students. Contents as followed.

‘Engineering disciplines cover many subspecialties, including but not limited to civil engineering, electrical engineering, chemical engineering, etc. Most branch majors study some general subjects. Such as algebra, statistics, chemistry, and physics. The study plans and study purposes are different between different branches, which leads to the different subjects they study. For example, civil engineering students will study subjects including but not limited to hydraulic engineering, structural engineering, building materials, foundation engineering, etc.’

1.4 Take care of the orderly numbering of section 2.4 Measurements.

Response: Thank you for your suggestion. We already revised it and highlighted it.

1.5 To deepen the implications of the university in health and social welfare programmes.  See

Lázaro-Pérez, C.;Gómez, P.M.;  Martínez-López, J.Á.;Gómez-Galán, J. Predictive Factors ofSuicidal Ideation in SpanishUniversity Students: A Health,Preventive, Social, and CulturalApproach.  J. Clin.  Med. 2023, 12, 1207.https://doi.org/10.3390/jcm12031207

Response: Thank you for your suggestion. It’s a good guidance for us to consider the relationship in the results we got from our study between university and society, we added some information in ‘Implication and future research’ section based on your comments as followed.

A more comprehensive approach should be established to enhance early detection and intervention for mental health problems among university students(Lázaro-Pérez et al., 2023). Universities should implement prevention and early intervention programs through their healthcare facilities. These programs could focus on developing interpersonal and social skills among students, as well as initiatives that foster social connectedness and bolster self-esteem and self-efficacy. Early detection is crucial in identifying mental health issues before they escalate, and universities play a vital role in creating a supportive environment for students. By offering proactive programs that address various aspects of mental well-being, such as social skills, social support, and self-confidence, students can be better equipped to cope with the challenges they face during their academic journey. These initiatives could involve workshops, counseling services, peer support groups, and awareness campaigns to raise mental health awareness on campus.

Reviewer 2
2.1 The Introduction section needs revision. There are a lot of variables in the study and authors should be an accurate literature revision for all of them. Moreover, some sentences are incomplete and the ideas/arguments not completely explained. For example, p.2, lines 60-61.” Researchers found that depression among engineering students were associated with male gender, internet use [8, 9], and lacking social skills[7], and alcohol use disorder[10].” I wonder: In which way there are these associations? Moreover, I believe that it would add value to the manuscript to include cultural lens. In this sense, Asian context has sociocultural features that differ from America and Europe. I recommend authors to take this point into consideration in order to modificated the literature review. Finally, I recommend authors include a section “the present study”. This section should be in another paragraph. Please, make sure all objectives are stated in this section and explain the results expected.

Response. Thank you for your suggestion. We have modified the lines 60-61 as followed.

‘Researchers found that depression among engineering students was associated with internet addiction and the frequency of internet use. High frequency of internet use and internet addiction were positively correlated with depression among engineering students, and social skills were negatively correlated with clinical depression, and alcohol use was a strong predictor of depression. Individuals who start using alcohol as underage are more likely to become addicted than individuals who stated at age 20 or over.’

And yes, your suggestions about cultural lens are very insightful. We have added related contents in ‘Introduction’ section, added information as followed.

Additionally, culture can play a significant role in shaping mental health problems. Traditional East Asian culture, for example, places a strong emphasis on collectivism, in contrast to the individualism commonly found in Western culture. While Asian culture may exhibit stigma around mental health and limited emotional expression, it also values familial and social support(Ariapooran et al., 2018; Kim & Jang, 2019). Students from collectivistic cultures might face heightened academic and societal pressure. Furthermore, the expression of mental health problems, including depression, can be notably influenced by cultural factors(Dreher et al., 2017; Ronningstam et al., 2018; Snowdon, 2018).

We have added ‘the present study’ to explain objectives and the expective results of the study, contents as followed.

The objective of this study was to examine the prevalence of depression among undergraduate engineering students in Thailand and identify potential associated factors. The investigation encompassed sociodemographic data, learning styles, social skills, and various psychosocial factors. Building upon existing literature on university students, we anticipated that depression would also be prevalent among engineering students. Our hypothesis focused on certain psychosocial variables, such as low self-esteem and neuroticism, as significant predictors of depression in this population. Additionally, we aimed to compare alcohol use prevalence between engineering students and health science students. Furthermore, we explored the distinctive learning styles preferred by engineering students, though we did not formulate a specific hypothesis regarding the association between learning styles and depression. Ultimately, this study aimed to contribute to the understanding of mental health among engineering students and provide valuable insights for developing targeted interventions and support programs.

2.2   For all measures used, please indicate whether response scores were summed or averaged to create their composite scores for data analysis and add an example item for every scale or subscale used.

Response. Thank you for your suggestion. We have added description of each scale about total (summed) score for data analysis and we highlighted it in the ‘Measurements’ section.  The example item for each scale was displayed in the original ‘Measurements’ section.

2.3 I believe that it would add value to the manuscript to include two measures times in the analyses, a cross-sectional study is very descriptive and not allow to explore deeper the study objective proposed.

Response. Thank you for your constructive suggestion. We totally agree with that. However, as it is the first study among this population in Thailand, we need this survey to give us overall picture of how and what variables would play a role in their mental health. That’s why a simple study with cross-sectional design was created. However, we agree that multiple assessments should provide us with a more robust conclusion.   

For now, we have added additional information in the ‘limitation’ section.

Lastly, the study is a cross-sectional study, it’s difficult to determine a causal relationship between the outcome and predictive variables, longitudinal study is considered in the future study.’

2.4 Please follow my recommendations in the Introduction section and modify the discussion section accordingly. Once more, I believe that it would add value to the manuscript and that it would be very interesting to include a cultural lens in the discussion to interpret the results found. Maybe some results could be explained because of cultural effect?

Response. Thank you for your suggestion, we agreed with your comments, we have added some additional information in the ‘Discussion’ section, added contents as followed.

Despite the high prevalence rate, determining the extent of students' access to healthcare is challenging. Several factors, including cultural stigmatization, contribute to this difficulty(Augsberger et al., 2015). Similar to many East Asian cultures, students facing mental health issues may be reluctant to seek help or openly discuss their struggles.

Cultural factors may also contribute to limited emotional expression and socially inhibited interpersonal behaviors, which can impact social skills and eventually lead to depression. This effect becomes particularly notable during the COVID-19 pandemic, as social isolation exacerbates the situation(Gao et al., 2022; Lee et al., 2010).’

2.5 Please, revise limitations of the study. The participants of the present study are a very particular population (Asian population). Thus, this aspect should be a limitation of the study in order to avoid to generalize the results to all population. Another limitation is that it is a cross-sectional study, longitudinal studies being recommended in the future.
Response. Thank you for suggestion. We have added the longitudinal study statements in the future research and additional limitation information based on your comments, contents as followed.

Future research ‘Conducting longitudinal studies can help establish the causal relationship between these predictive factors and depression. Long-term investigations can provide valuable insights into how changes in these strong predictive factors over time affect the development and course of depression.’

LimitationAs this study was conducted on Thai students, it may not be applicable to engineering students from diverse cultural backgrounds.

2.6 I recommend authors to mention the cultural lens also in the conclusion section.

Response. Thank you for your comments, we have added some contents related to your suggestion, information as followed.

Thai engineering students may share similar subjects and study environments with foreign students, but they could be influenced by the culture of collectivism. This cultural aspect can impact their mental health, particularly leading to socially inhibited interpersonal behaviors. These behaviors, in turn, may make individuals more susceptible to depression, especially when facing academic or relationship pressures.

Reviewer 3

3.1 There are some minor typing errors such as in lines 21 and 29. Also it is best to be consistent with the gap between reference number and the last word of the sentences.

Response. Thank you for your suggestion. We appreciate it. In response, we have revised the typing problems that you pointed out, and we rechecked the whole paper and modified the gap between reference number and the last word of the sentences.

3.2 In the results, I would recommend having column heads in bold in Table 2 and Table 3. In Table 3, asterisks are not in the same format and that would be great if you can add the explanation for asterisks as a footnote.

Response. Thank you for this suggestion. We have revised the column heads in bold in Table 2 and Table 3. For the asterisks in Table 2, we revised it as the same format and added a footnote for it, revisions are highlighted.

3.3 In the discussion, the rate of depression is shared however I couldn't see that in the results section. I would recommend sharing it in the results first. In line 263, the reference should be at the end of the sentence. In the second paragraph of the discussion, it is mentioned that the depression rate is high in engineering students however compared to which group. It needs a reference. Because also in the abstract's conclusion part, the term unexpected high depression is used, however, it is not clear why unexpected, compared to what. I would suggest more references in that section and revision of the abstract's conclusion.

Response. Thank you for this suggestion. We have added the detail information of prevalence of depression in the results of Table 1 and we revised the citation problem in line 263. In the second paragraph of ‘Discussion’ section, we considered the high prevalence of depression as one of the important reasons to addressing of mental health issues among our target population. Your suggestion is in good insights, we modified the ‘Abstract’ section and contents in the ‘Discussion’ section as followed.

In abstract ‘The prevalence of depression among engineering students in Thailand was unexpectedly high compared with the prevalence of depression among engineering students in some countries.’

In discussion ‘The higher prevalence rates of depression (35.3%) among engineering students in this study among Thai engineering undergraduates compared with the prevalence of depression in two Turkish papers which conducted among Turkish engineering students (20.4% and 26.2% respectively) highlight the importance of addressing mental health issues in this specific population.’

3.4 In line 298 reference numbers are after the dot.

Response. Thank you for this point. We have revised and highlighted it.

3.5 Line 307, neuroticism is wide accepted predictor for depression, needs a reference.

Response. Thank you for this suggestion. We have added references to support the point of view and highlighted it as references [63-65].

We hope we are able to address the editor and the reviewers’ concerns. Please let us know if we need to improve our manuscript further. We are looking forward to hearing from you soon.

Best regards,

YH, TW and colleagues

Reviewer 2 Report

Based on a sample of 346 participants, this this paper analyzes the prevalence of depression and its associated factors among engineering students.

 ---Specific comments---

1.      The Introduction section needs revision. There are a lot of variables in the study and authors should be an accurate literature revision for all of them. Moreover, some sentences are incomplete and the ideas/arguments not completely explained. For example, p.2, lines 60-61.” Researchers found that depression among engineering students were associated with male gender, internet use [8, 9], and lacking social skills[7], and alcohol use disorder[10].” I wonder: In which way there are these associations? Moreover, I believe that it would add value to the manuscript to include cultural lens. In this sense, Asian context has sociocultural features that differ from America and Europe. I recommend authors to take this point into consideration in order to modificated the literature review. Finally, I recommend authors include a section “the present study”. This section should be in another paragraph. Please, make sure all objectives are stated in this section and explain the results expected.

2.      For all measures used, please indicate whether response scores were summed or averaged to create their composite scores for data analysis and add an example item for every scale or subscale used.

3.      I believe that it would add value to the manuscript to include two measures times in the analyses, a cross-sectional study is very descriptive and not allow to explore deeper the study objective proposed.

4.      Please follow my recommendations in the Introduction section and modify the discussion section accordingly. Once more, I believe that it would add value to the manuscript and that it would be very interesting to include a cultural lens in the discussion to interpret the results found. Maybe some results could be explained because of cultural effect?

5.      Please, revise limitations of the study. The participants of the present study are a very particular population (Asian population). Thus, this aspect should be a limitation of the study in order to avoid to generalize the results to all population. Another limitation is that it is a cross-sectional study, longitudinal studies being recommended in the future.

6.      I recommend authors to mention the cultural lens also in the conclusion section.

Author Response

(The authors gave the same response as above.)

Reviewer 3 Report

Great work! I have some minor comments.

1. There are some minor typing errors such as in lines 21 and 29. Also it is best to be consistent with the gap between reference number and the last word of the sentences. 

2. In the results, I would recommend having column heads in bold in Table 2 and Table 3. In Table 3, asterisks are not in the same format and that would be great if you can add the explanation for asterisks as a footnote.

3. In the discussion, the rate of depression is shared however I couldn't see that in the results section. I would recommend sharing it in the results first. In line 263, the reference should be at the end of the sentence. In the second paragraph of the discussion, it is mentioned that the depression rate is high in engineering students however compared to which group. It needs a reference. Because also in the abstract's conclusion part, the term unexpecdey high depression is used, however, it is not clear why unexpected, compared to what. I would suggest more references in that section and revision of the abstract's conclusion.

4. In line 298 reference numbers are after the dot.

5. Line 307, neuroticism is wide accepted predictor for depression, needs a reference.

Author Response

(The authors gave the same response as above.)
